# RIoT: A Large-Scale Real-World IoT Dataset

## Abstract

Despite the rapid proliferation of IoT devices, research progress is hindered by the lack of large-scale, publicly available datasets. Existing resources are often synthetic, short-lived, or overly simplistic, limiting their value for building realistic IoT systems. To address this gap, we present RIoT, a comprehensive real-world IoT dataset collected from six diverse building deployments over a cumulative six-year period. RIoT spans more than 200 sensors and over 10 million measurements across multiple modalities, including HVAC usage, energy consumption, and air quality, and captured at high temporal resolution with semantic and spatial annotations. By coupling scale with context, RIoT provides a realistic foundation for advancing IoT research, enabling the community to move from prototype-level studies toward robust, deployable systems.

## 1 Introduction

Despite the explosive growth of the Internet of Things (IoT) (Madakam et al., 2015), its true promise, namely collecting data from edge devices and using it to enable "smart" decision-making (Mahmoud et al., 2015), remains far from fully realized. Most IoT devices in practice continue to operate primarily as passive observers (Subahi & Theodorakopoulos, 2019), with the expectation that they will eventually evolve into active participants (Sateeschchandra et al., 2025). Meanwhile, IoT adoption continues to proliferate, demonstrated by the increase in the number of connected devices rising from 7 billion in 2019 to 14 billion in 2023 (Vailshery, 2024) to potentially exceeding 32 billion by 2030 (Vailshery et al., 2024). This growing scale underscores the importance of advancing research into IoT systems to overcome their known challenges (Haroon et al., 2016) and move beyond passive cloud-connectivity (Pereira et al., 2013) toward active applications that close the sensing-actuation loop (Ikuabe et al., 2020). For instance, as global energy demand surges (Zohuri, 2023; Nassar & Khaleel, 2024), IoT-enabled systems could play a critical role in optimizing the energy needed to illuminate, heat, cool, and ventilate buildings (Minoli et al., 2017).

A major challenge holding back IoT research is the *chronic lack of access to large-scale, real-world IoT data* (Mansouri et al., 2021; Haroon et al., 2016; Bouke et al., 2024; Bansal et al., 2020; Khare & Totaro, 2019). Companies are usually reluctant to share such data (Jernigan et al., 2016), commonly for privacy (Stach et al., 2022) and security (Harper et al., 2022) reasons, or vendor lock-in (van der Zeeuw et al., 2024) incentives. For example, while modern buildings often have hundreds of IoT sensors that measure temperature, power consumption, air quality, etc., access to real-world building data remains limited (Wang et al., 2022). As a result, *Smart Buildings* [1] (Jia et al., 2019; Brad & Murar, 2014; Mocrii et al., 2018; Paul et al., 2018; Samuel, 2016b) research is often forced to resort to synthetic or controlled laboratory data, which ultimately hinders the development of realistic solutions. Looking ahead, this scarcity is unlikely to improve; beyond existing concerns, companies are increasingly viewing IoT data as a strategic asset, particularly valuable for training proprietary ML models (Song et al., 2018), which further reduces incentives to share.

Specifically, a closer look at the current publicly available IoT datasets (Table 1) reveals several limitations that constrain their utility: they are (❶) often collected in laboratory settings or controlled testbeds, or (❷) heavily specialized in ways that offer only narrow perspectives and hinder causal understanding. For instance, even real-world datasets beyond lab-created environments frequently track only single variables, without accompanying context like spatial information, HVAC [2] activ-

---

[1] alongside its industry counterpart, Building Management Systems (BMS) (Manic et al., 2016)
[2] HVAC stands for Heating, Ventilation, and Air Conditioning.

Table 1: Comparison between existing IoT datasets and `RIoT` across representative dimensions: multi-site coverage, spatial tagging, indoor/outdoor sensing, multi-modality, and recording duration. Most prior datasets are limited in scope, typically single site, short-term, and focused on single sensor modalities. In contrast, `RIoT` offers recordings across multiple diverse deployments, with complete spatial annotations across heterogeneous sensor types, making it a significantly more realistic and comprehensive resource.

| Work | Multi-site | Spatial Inf. | Ind.& Out. | Multi-mod. | Duration |
|---|---|---|---|---|---|
| **RIoT (our work)** | ✔ | ✔ | ✔ | ✔ | 18 months |
| TSDP (Kumar et al., 2020) | ✗ | ✗ | ✗ | ✗ | 3 months |
| Tapia (Tapia et al., 2004) | ✔ | ✔ | ✗ | ✗ | 0.5 months |
| AARHS (van Kasteren et al., 2008) | ✔ | ✔ | ✗ | ✗ | 1 month |
| DLABAC (Mocanu et al., 2016) | ✗ | ✗ | ✗ | ✗ | 47 months |
| BLUED (Filip et al., 2011) | ✗ | ✗ | ✗ | ✗ | 0.25 months |
| Powersmith (pow, 2018) | ✗ | ✗ | ✗ | ✗ | 24 months |
| CU-BEMS (Chitalia et al., 2020) | ✗ | ✗ | ✗ | ✗ | 11 months |
| BDGP2 (Miller et al., 2020) | ✔ | ✗ | ✗ | ✗ | 24 months |

ity, occupancy, or similar metrics that could only collectively provide a comprehensive picture of a particular site and its measurements in context of human, environmental, and operational impacts.

First (❶), although such settings offer clean and well-instrumented data valuable for early-stage research, they often fail to reflect the diversity of real-world patterns across different physical contexts. For instance, Figure 1a shows an example of real-world measurement variability: while all displayed temperature sensors are placed on the same floor of a building, their readings vary substantially (the average deviation across sensors at each time point exceeds 10°C). The contrast becomes starker in Figure 1b, where outdoor sensors reveal additional measurement dynamics, pool water remains stable (3°C deviation) while outdoor temperature exhibits sharp diurnal swings (12°C deviation). Consequently, as physical context, function, and microclimate fundamentally shape sensor behavior, such distinctions cannot be captured in synthetic datasets, which in turn limits progress towards the development and research of realistic use-cases.

Second (❷), even when datasets span longer timeframes or involve larger deployments, they tend to be vertically focused, centered around single data modalities such as energy metering, HVAC activity (Gholamzadehmir et al., 2020), or network traffic (Kumar et al., 2021), without integration across domains. For instance, while datasets like BDGP2 (Miller et al., 2020) contain millions of power metering records, leveraging them beyond narrow use cases such as energy consumption modeling is not possible; ideally, such datasets would also include different contextual signals like temperature, occupancy, or equipment schedules to support causal mod-

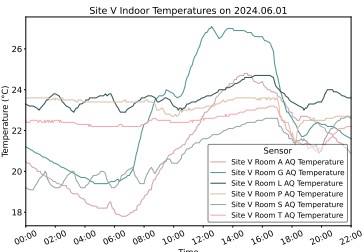

(a) Diverse measurement patterns among sensors placed inside building

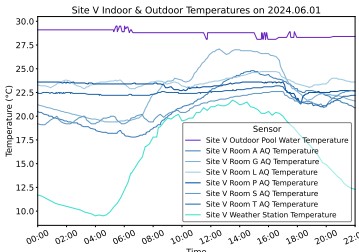

(b) Differences in measurement patterns among sensors placed inside (colored in shades of blue) vs. outside (purple and turquoise lines)

Figure 1: Example of real-world measurement pattern diversity. Measurements recorded at `Site V`: (a) differences among sensors located inside the building, and (b) differences between sensors located both inside and outside.

eling. The reality, however, is that it is rarely the case for datasets to contain sensors' operational context, with such information frequently missing: sensors are rarely annotated with their physical location within sites, limiting the ability to model interdependencies (more details about specific

limitations of in § 5). Without sufficient environmental context, the data remains observational, lacking the information necessary to close the sense-actuate loop (Ikuabe et al., 2020).

To address these limitations, we release a large-scale **R**eal-world **IoT** dataset—**RIoT**— aiming to empower researchers to build, test, and iterate IoT solutions grounded in real-world conditions. The published dataset is comprised of anonymized, proprietary IoT data obtained in collaboration with an industry-leading BMS company. Unlike previous datasets, RIoT is multi-dimensional, multi-site, and also spatially tagged through the adoption of a unified tagging scheme (John et al., 2020). In summary, we make the following contributions:

- We introduce the first comprehensive, large-scale real-world IoT dataset, RIoT, which contains over 10 million sensor measurements (two orders of magnitude higer compared to similar datasets), across six deployment sites.
- We analyze RIoT and visualize the measurement distributions across all sensor types to clearly quantify available resources.
- We demonstrate two example use-cases leveraging RIoT: (1) anomaly detection for firmware bugs and detection of rate-change sensor behavior for indoor/outdoor environments and (2) HVAC optimization through occupancy modeling.

Consequently, RIoT's goal is to provide a foundation the community can build on, to move beyond discussions about what IoT could be, and start demonstrating what it can now deliver. The dataset and all the code for experiments used in this article can be accessed at https://anonymous.4open.science/r/RIoT-9798.

## 2 DATASET CHARACTERISTICS AND COLLECTION

**Data Collection and Polling Intervals.** RIoT was collected from six different sites (Table 2), deploying various types of sensors, measuring parameters like temperature, humidity, $CO_2$, energy consumption, and other environmental or operational indicators (full list of sensor types in Table 3). The deployments differ in scale and duration, due to the incremental rollout of IoT sensors across the locations (visualization in Figure 4). For instance, Site M and Site S1, each monitored over a full 18-month period with 68 and 41 sensors respectively, contribute substantially to the overall volume, generating 774 MB and 729 MB of data. By comparison, deployments such as Site J and Site S2—with approximately 10 sensors each—expectedly generate smaller data volumes (60.8 MB and 49.6 MB respectively).

Table 2: Data recording durations and volumes per site. Cutoff collection date is 01-09-2025. "cu." = cumulative.

| Site no. | Site name | Duration (months) | No. sensors | Total size | Start date | No. floors | No. rooms |
|---|---|---|---|---|---|---|---|
| 1 | Site V | 18 | 27 | 502 MB | 31-03-2024 | 3 | 8 |
| 2 | Site S1 | 18 | 41 | 729 MB | 01-04-2024 | 3 | 8 |
| 3 | Site M | 18 | 68 | 774 MB | 20-04-2024 | 5 | 17 |
| 4 | Site S2 | 10 | 8 | 49.6 MB | 16-08-2024 | 1 | 2 |
| 5 | Site J | 10 | 10 | 60.8 MB | 23-08-2024 | 1 | 2 |
| 6 | Site T | 8 | 74 | 362.1 MB | 18-12-2024 | 4 | 25 |
| *Total* | – | cu. 82 (6.8 years) | 228 | 2.42 GB | – | 17 | 62 |

All installed sensors (228 across all sites) communicate recorded measurements using Lo-RaWAN (de Carvalho Silva et al., 2017), which provides long-range, low-power wireless transmission to a central gateway (one per site) that periodically stores the collected data. Each sensor operates with a polling interval of 15 minutes, regularly recording its measurements. Specifically, all sensors are configured with an *on-change* policy setting: if the sensed value changes by more than 1 unit of measurement, e.g., a 1° change in temperature or a 1 ppm change in $CO_2$, the sensor immediately records the new reading and broadcasts it to the gateway. This dual-mode reporting (combining periodic polling with threshold-driven updates) is essential to accurately reflect real-world dynamics, and allows the dataset to capture both stable trends and sudden fluctuations. Consequently, compared to publicly available IoT datasets, which rely on fixed sampling intervals, RIoT offers a new level of temporal resolution (compared to RIoT's default 15 minute interval,

typical recording windows in other datasets are 1 hour intervals (Ansari & Alam, 2024), especially over longer durations, i.e., spanning months), which makes it especially valuable for applications that require timely detection of rapid environmental changes. We showcase the value of the *on-change* policy in § 2, where we explore modeling sensor rates of change in different indoor/outdoor environments.

**Multi-site deployment as a resource for application prototyping.** Figure 2 showcases data accumulates over time: larger sites with higher sensor counts, while deployed later (e.g, `Site T`), accumulate data faster, compared to other smaller deployments generate lighter data streams (e.g, `Site S2`). This diversity enables researchers match sites to application needs, from larger, multi-floor environments, suited for more complex modeling and control, to more compact deployments, ideal for lightweight or single-digit sensor analyses.

**Spatial awareness for sensors through tagging.** Crucially, we adopted the Haystack tagging scheme (John et al., 2020), an *open standard* for describing building equipment and sensor data using semantic tags. This fundamentally enables a better way to understand the sensors in context, alleviating the challenges associated with data integration (Minoli et al., 2017) and ecosystem interoperability (Samuel,

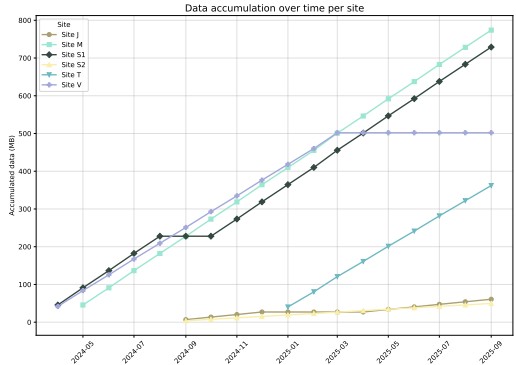

Figure 2: Data accumulation over time per site; larger, sensor-dense sites generate more data, while smaller deployments produce lighter volumes, enabling selection of sites tailored to specific application scenarios.

2016a), typically encountered in BMS systems. This stands in stark contrast to traditional approaches, such as KNX (Martirano & Mitolo, 2020), which typically operate at the level of Modbus registers (Peng et al., 2008), handling data as low-level numerical values without any semantic context, resulting in datasets that either lack the context required for interpretation or omit tagging entirely. By systematically labeling entities at installation time by design (e.g., `tempSensor`, `phSensor`, `energyMeter`) and linking them to site references (`siteRef`), as well as to `floor`, and `room` metadata, heterogeneous data sources can be readily contextualized and integrated.

**Preprocessing and anonymization.** In order to protect the privacy of the site owners and their employees, the dataset was anonymized with care, ensuring that sensitive information would not be leaked while preserving, as much as possible, properties such as geolocation that could be vital for contextualizing the dataset by considering other, external resources such as historical weather information or outdoor humidity data. To achieve that, the geolocations contained in the raw data were obfuscated through generalization and pertubation, adding a potential inaccuracy of up to 20km in radius. We consider this to be a reasonable choice, given that, e.g., in the UK where three of the sites are located (`Site M, J, and T`), the average distance between MetOffice weather stations is 40km (UK MetOffice).

**Sensor data types and characteristics.** Compared to current datasets, which typically record a relatively limited number of sensor modalities, `RIoT` supports an extensive array of sensor measurement types, including air quality monitoring (e.g., $CO_2$, PM levels, TVOC, humidity), odor detection (ammonia, hydrogen sulfide), climate control (motor position and stroke), energy monitoring (current and total current), weather tracking (temperature, pressure, wind), and water level and temperature sensing.[3].

**Data accumulation and availability periods.** The overall distribution of measurements and their storage requirements is summarized in Figure 3, where (a) shows the number of measurements per sensor type with corresponding storage footprints and (b) illustrates how these measurements accumulate over time per sensor type. The underlying storage assumptions are detailed in Table 4, where

---

[3]This is another dimension in which `RIoT` may be of potential value: real-world deployments that track sensor data over extended periods of time will inevitably experience technical failures. Training on data that includes such imperfections, such as outliers and missing values, is crucial for developing robust AI solutions. The repository also includes fine-grained ingestion rates and sensor coverage information.

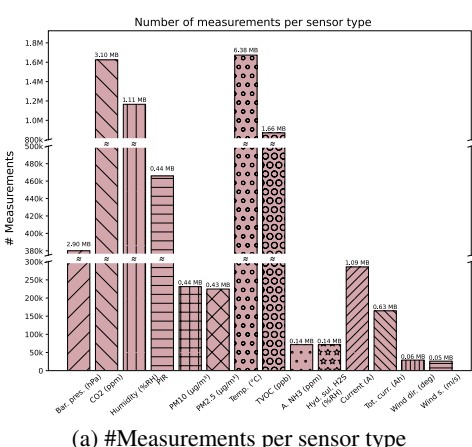

(a) #Measurements per sensor type

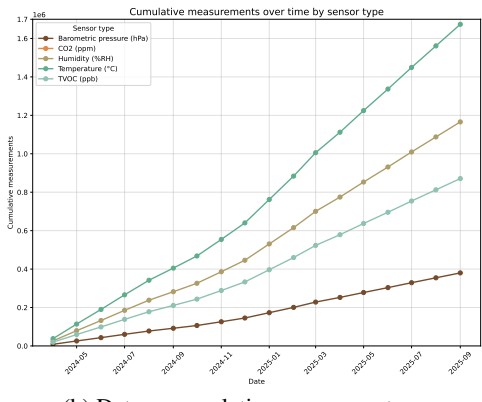

(b) Data accumulation per sensor type

Figure 3: Estimates for no. measurements with corresponding data volumes as an estimate for resources available along different sensor verticals. Specifically, (a) # measurements across sensor type with data volume estimates and (b) measurement accumulation per sensor type.

Table 3: `RIoT`'s sensor composition. Cutoff collection date is `01-09-2025`. Cumulative duration was calculated by adding up all recorded intervals across all sensors of that particular type. "C. dur." = Cumulative duration.

| Sensor type | #Sensors | C.dur. (hours) | #Measurements | C.dur. (years) | Earliest recording |
|---|---|---|---|---|---|
| Barometric pressure (hPa) | 14 | 17 839 | 380 030 | 2.03 | 31-03-2024 |
| $CO_2$ (ppm) | 21 | 27 984 | 1 624 601 | 3.19 | 31-03-2024 |
| Humidity (%RH) | 54 | 81 606 | 1 165 799 | 9.31 | 31-03-2024 |
| PIR | 13 | 15 369 | 465 951 | 1.75 | 31-03-2024 |
| PM10 (µg/m³) | 3 | 5 184 | 231 342 | 0.59 | 31-03-2024 |
| PM2.5 (µg/m³) | 3 | 5 184 | 224 761 | 0.59 | 31-03-2024 |
| Temperature (°C) | 57 | 86 318 | 1 673 692 | 9.85 | 31-03-2024 |
| TVOC (ppb) | 13 | 15 369 | 870 930 | 1.75 | 31-03-2024 |
| Ammonia NH3 (ppm) | 3 | 5 842 | 71 468 | 0.66 | 31-03-2024 |
| Hydrogen sulfide H2S (%RH) | 3 | 5 842 | 71 473 | 0.66 | 31-03-2024 |
| Current (A) | 21 | 8 066 | 285 461 | 0.92 | 14-05-2024 |
| Total current (Ah) | 21 | 8 183 | 164 694 | 0.93 | 14-05-2024 |
| Wind direction (deg) | 1 | 2 472 | 29 072 | 0.28 | 10-06-2024 |
| Wind speed (m/s) | 1 | 2 472 | 26 204 | 0.28 | 10-06-2024 |
| *Total* | 228 | 287 730 | 10 226 204 | 32.79 | – |

Table 4: Sample measurements and estimated storage requirements per record. Datatypes were chosen based on observed ranges and precision.

| Measurement type | Sample value | Bytes per measurement |
|---|---|---|
| Barometric pressure (hPa) | 988.2999877929688 | 8 |
| $CO_2$ (ppm) | 502 | 2 |
| Humidity (%RH) | 58 | 1 |
| Temperature (°C) | 24.200000762939453 | 4 |
| PM10 (µg/m³) | 6 | 2 |
| TVOC (ppb) | 279 | 2 |

datatypes were chosen based on observed ranges and precision (e.g., humidity stored as `uint8`, temperature as `float32`)[4]. Together with the availability timeline in Figure 4, these views highlight both the long-term accumulation of data across diverse sensor modalities and the operational realities of downtime, which naturally introduce gaps into the dataset.

---

[4]For space limitation, we provide samples for the first seven most frequent measurement types. A complete specification of all measurement types is available in the repository published alongside this work.

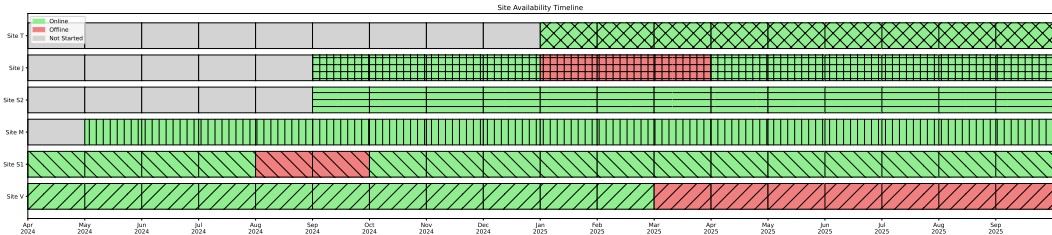

Figure 4: Availability timeline per site. Light green marks periods when sensors were online; light red marks offline periods. These availability gaps arise naturally in long-running deployments (e.g., network issues or hardware downtime), highlighting operational realities captured.

Figure 4 shows the availability timeline per site: periods of downtime (in red) are visible across sites, and these gaps mainly stem from gateway-level downtime. In those cases, particularly for `Site J`, `S1`, and `V`, the reason for unavailability stems from the fact that the gateways were offline (due to maintenance or other reasons) and could not collect and record the data from the sensors.

# 3 USE CASES

Most prior work presents use-cases such as forecasting (Ansari & Alam, 2024) or anomaly detection (Nizam et al., 2022) as stand-alone analytics on curated streams. These are valuable for improving prediction accuracy for changes in particular sensor modalities, e.g., for temperature, humidity, etc., but they rarely serve as efforts for exploring end-to-end deployments in which sensed data must ultimately drive decisions and actions. Consequently, the use-cases considered in this study have the goal to showcase how `RIoT` can bridge this gap, while acknowledging that both modes matter: narrow, single-stream tasks remain important for smaller-scale applications, whereas full-loop scenarios are essential to realize IoT's promise. In Figure 5, we depict the broader architecture of modern IoT systems and highlight the steps that are typically missing in practice (⑦ and ⑧). We stress that the purpose of introducing `RIoT` is to serve as a resource for building the missing link, as this gap is arguably not due to a lack of conceptual understanding (Zhang & Tao, 2021), but rather due to the scarcity of realistic IoT datasets needed for development and evaluation.

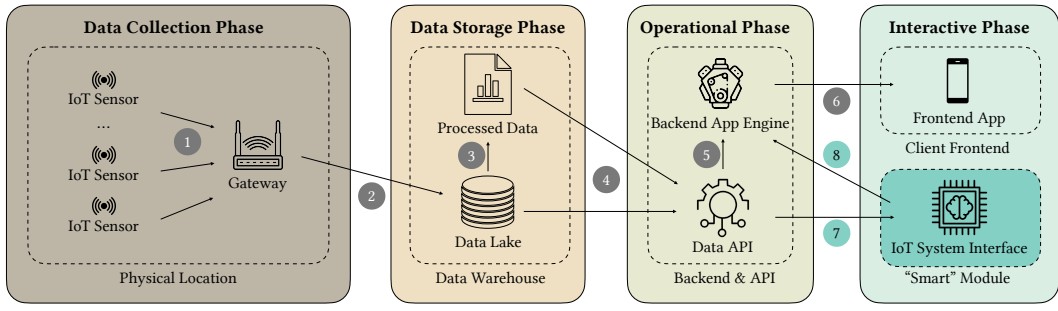

Figure 5: Typical stages of an IoT deployment. ① IoT sensors send measurements to a gateway, which subsequently forwards it to a warehouse (②). For efficiency, raw sensor data is often transformed into usable data points (step ③), e.g., through ETL processes (Simitsis et al., 2005), made available for API consumption (④), and finally consumed by backend engines (⑤) that present it to some form of user front-end application (⑥). Steps ⑦ and ⑧, arguably the most important steps for closing the sense-actuate loop, are typically absent from practical deployments. `RIoT` is intended to address this gap by providing the data necessary for prototyping solutions.

Concretely, we show three example use-cases. First, we explore anomaly detection in two forms that suit single-stream settings (§ 3.1): firmware-induced outliers that are syntactically valid yet physically implausible, and rate-of-change anomalies that contrast smooth indoor dynamics with inherently more volatile outdoor signals. Second, we present HVAC optimization through occupancy

modeling, where $CO_2$ (corroborated by TVOC) is used to derive a per-room control signal that can stop or reduce heating while maintaining comfort, turning sensing into actuation (§ 3.2).

## 3.1 ANOMALY DETECTION: FIRMARE BUGS AND RATE-OF-CHANGE IN INDOOR/OUTDOOR ENVIRONMENTS

**Firmware induced anomalies.** Real-world IoT deployments are heterogeneous by design, as they comprise a multitude of sensors (originating from different vendors), which increases the chances of systematic fleet-wide faults, that look syntactically valid but are semantically wrong.

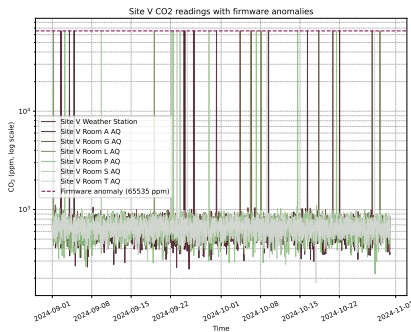

(a) Firmware induced $CO_2$ anomalies at `Site V`.

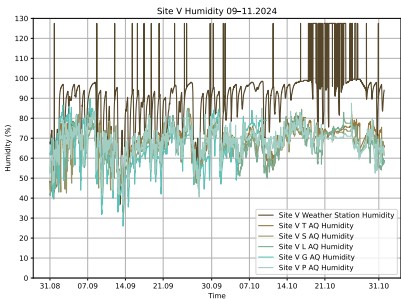

(b) Humidity dynamics at `Site V` without firmware errors.

Figure 6: Measurements in (a) firmware artifacts manifesting as extreme but systematic outliers, and (b) natural variability without firmware faults.

As a concrete example, during the curation of `RIoT`, we identified a repeatable $CO_2$ (`ppm`) firmware artifact: due to a sign handling error, some devices occasionally reported the maximum register value of the measurement slot (65535ppm). Such a reading is physically implausible indoors[5] and we labeled every such line as *anomaly*, with all other readings labeled *normal*[6]. Across all $CO_2$ (1,767,652 measurements in total), 490 were such anomalies (0.277%, similar to relative amounts of imbalance found in other works (Giannoni et al., 2018)). As shown in Fig. 6a, these firmware artifacts manifest as extreme outliers: while normal indoor $CO_2$ levels average approx. 500 ppm, the anomaly values spike to 65,535 ppm (over 130× larger). In contrast, Fig. 6b illustrates humidity dynamics at the same site, where values fluctuate between 50–80% (with occasional noise), but remain within a physically meaningful range without firmware errors.

**Rate of change anomalies in indoor vs. outdoor environments.** Most indoor physical variables (temperature, $CO_2$, TVOC, etc.) evolve gradually, while abrupt, large relative changes over short intervals are rare and often operationally meaningful, typically associated with sensor glitches or other anomalous behavior. In contrast, outdoor variables are naturally more volatile, driven by weather and diurnal cycles, and thus exhibit much broader distributions of change. As shown in Figure 1b, indoor temperature sensors (shades of blue) evolve smoothly within a narrow band, while the outdoor weather station (turquoise) displays pronounced swings following diurnal heating and cooling. However, some outdoor signals such as the pool water temperature (purple) remain nearly constant due to thermal inertia, highlighting that outdoor variability is modality-dependent.

This contrast leads to one key insight: *indoors, abrupt relative changes are strong indicators of anomalies*, while outdoors rates of change vary more widely by sensor type and environmental exposure. As such, `RIoT` is the ideal resources for this distinction explicit: thanks to its semantic spatial tagging, which clearly differentiates indoor from outdoor deployments, and its multi-modal sensor coverage, we can systematically compare how change rates behave across contexts. What is

---

[5]Such a concentration is more than an order of magnitude above accepted occupational exposure limits: the EU *level limit value* (LLV; 8-hour TWA) for $CO_2$ is 5000 ppm (Af Petersens et al., 2020).

[6]We flag these anomalies in a separate section of the dataset, to enable future work to directly use it for anomaly detection.

key, however, is that the dataset *on-change* policy ensures that both smooth trends (typical of indoor environments) and sudden excursions (e.g., outdoor weather swings or sensor faults) are preserved.

**Evaluation.** To quantitatively assess anomaly detection, we trained lightweight detectors (z-score, seasonal-hybrid extreme studentized deviation (Vieira et al., 2018), and an LSTM (Wei et al., 2023)) on the two scenarios. For the firmware case, all detectors achieved perfect separability, as the outliers are extreme and physically implausible. In contrast, rate-of-change anomalies were subtler (AUROC

Table 5: Baseline anomaly detection performance. Firmware anomalies are trivially separable, while rate-of-change anomalies require fine-grained dynamics captured by `RIoT`'s on-change policy.

| Method | Firmware ($CO_2$) | Rate-of-change |
|--------|-------------------|----------------|
| Z-score | 1 | 0.83 |
| SH-ESD | 1 | 0.87 |
| LSTM | 1 | 0.91 |

=0.83–0.91), highlighting the importance of `RIoT`'s *on-change* policy: without fine-grained temporal resolution, such short-lived excursions cannot be observed or modeled. This distinguishes `RIoT` from existing datasets in the literature, which typically rely on coarser (1-hour intervals, see § 5), fixed sampling intervals and therefore cannot support detectors that rely on rapid environmental changes.

## 3.2 HVAC Optimization thorough Occupancy Modeling

**$CO_2$ and TVOC as proxies for occupancy.** Closing the sensing–actuation loop is where IoT systems move from passive data collection to active participants. Consequently, we now consider a use case that demonstrates how `RIoT`'s site–floor–room structure and multi-modal signals can be used to derive control signals for HVAC from implicit occupancy, thereby optimizing energy consumption. The intuition is simple: when rooms are used, $CO_2$ and TVOC rise (Marques et al., 2019); when they are empty, both remain constant and gradually decay. Because all sensor are semantically tagged to sites, floors, and rooms, signals can be paired with specific spaces and cross-validated across adjacent rooms.

**Evaluation.** We consider `Site T`, the largest site in the deployment with respect to number of rooms, i.e., 25 (Table 2). Out of the 25, 24 rooms are equipped with $CO_2$, TVOC, temperature, and humidity sensors. Among these, 5 are meeting rooms, which exhibit different occupancy dynamics than offices (for illustrative purposes, we consider a time-window threshold, namely if $CO_2$ and TVOC deltas exceed 100 ppm and ppb points for over 10 minutes, respectively; this is consistent with other studies (Vanus et al., 2019)). Consequently, $CO_2$ traces show that meeting rooms are typically in use only one day per week. Out of 40 potential heating hours per week (8 h × 5 days), they are occupied ∼8 h, i.e., 20% of the time. If left on continuously, each meeting room would consume ∼48 kWh per week. With occupancy-based control, this drops to ∼9.6 kWh, saving 80%. Over four months (January-April 2025, approximately 120 workdays), the five meeting rooms alone save ∼1,500 kWh. The same mechanism is applied to officies, where we observe they are in use roughly 50% of the time. Aggregating across both room types, `Site T` could save nearly 14,000 kWh over the four-month period. Table 6 summarizes baseline consumption and savings for meeting rooms and offices, illustrating that meeting rooms yield higher relative savings, while offices contribute more in absolute terms.

While this use case focuses on inferring occupancy from air-quality proxies, dedicated sensors such as PIR also play an important role. Although not always uniformly deployed, PIR provides valuable ground truth: with explicit occupancy labels, one could estimate the upper bound of an ideal HVAC policy, namely turning heating off immediately when a room is empty and restoring it upon presence. We

Table 6: Estimated HVAC energy savings at `Site T` over Jan–Apr 2025 (∼120 workdays). Meeting rooms show higher relative savings due to sparse usage; offices contribute more in absolute terms.

| Room type | # rooms | Baseline (kWh) | Proxy (kWh) |
|-----------|---------|----------------|-------------|
| Meeting | 5 | 1,920 | 384 |
| Offices | 19 | 20,900 | 8,800 |
| Total | 24 | 22,820 | 9,200 |

therefore consider a variant of the experiment where PIR signals are available. The actuation signal becomes more reliable, further improving energy savings. In practice, however, even without PIR, facilities can already benefit by leveraging proxy signals ($CO_2$, temperature, humidity, TVOC), which are nearly always present. Quantifying the performance delta between the two scenarios remains an interesting direction for future work.

## 4 LIMITATIONS AND FUTURE WORK

**Dataset scope.** Although RIoT spans six sites and more than ten sensor modalities, it remains focused on building-related deployments. Extending coverage to automotive or mobile IoT settings would further increase representativeness and broaden applicability.

**Data quality and reliability.** As with any real-world dataset, RIoT contains imperfections: missing values due to outages, and hardware-induced anomalies. While we tried to flag all imperfections (for instance, explicitly labeling the firmware-induced recordings), others may remain. We view these as a feature rather than a bug, since robustness to such imperfections is critical in practice. Future work could enrich the dataset with additional annotations to support supervised robustness studies.

## 5 RELATED WORK

**IoT datasets.** Table 1 summarized how existing datasets are particularly constrained by short duration, narrow sensing modalities, or by missing spatial and contextual information. These datasets often lack the temporal depth, spatial diversity, and sensor heterogeneity required to model real-world conditions accurately. For example, TSDP (Kumar et al., 2020) spans only 3 months and contains air quality data from three stations, while Tapia (Tapia et al., 2004) and AARHS (van Kasteren et al., 2008) rely on binary sensors across two buildings but cover only half a month and one month, respectively. Other datasets emphasize longer deployments but remain vertically narrow: DLABAC (Mocanu et al., 2016) (47 months) and BLUED (Filip et al., 2011) (one week, 47 sensors) focus exclusively on power metering, as do Powersmith (pow, 2018) (24 months) and CU-BEMS (Chitalia et al., 2020) (11 months). Even BDGP2 (Miller et al., 2020), which spans 24 months across multiple sites, is limited to power metering data and and omits spatial metadata about sensor placement within sites. Collectively, current datasets provide valuable but partial views of IoT ecosystems, motivating the need for multi-modal, spatially grounded resources such as RIoT.

**Outlier prediction and sensor modeling.** IoT research dates back to the 1980s (Madakam et al., 2015), with much of the literature focusing on analytics over curated, single-stream datasets (Sater & Hamza, 2021). Time-series forecasting has been widely studied, from ARIMA (Abdelmgeed et al., 2023) to deep learning (LSTMs (Zhang et al., 2018), Transformers such as Informer (Zhou et al., 2021) and TimesNet (Wu et al., 2023)). Benchmarks like M4 (Makridakis et al., 2018) and datasets such as UCI Air Quality (Vito, 2008) provide testing grounds but are typically small, synthetic (Alabdulwahab et al., 2023; Anderson et al., 2014), or highly feature-engineered. A large body of work also targets anomaly detection (Cook et al., 2019), identifying statistical outliers in sensor streams (Nizam et al., 2022). While effective in controlled settings, these approaches lack contextual grounding and ignore inter-sensor dependencies. By contrast, RIoT's spatial tagging enables reasoning about anomalies as system-level events rather than isolated irregularities.

**LLM-enabled IoT.** Recently, natural language foundation models (Bommasani et al., 2021) have sparked growing interest in interactive, autonomous IoT systems, with frameworks proposed for integrating language models into IoT pipelines (Cui et al., 2024). Yet current evaluations remain limited to synthetic or highly simplified datasets. As a future direction, it would be valuable to explore how RIoT could serve as a resource in such scenarios, given that it captures not only individual device signals but also full-deployment dynamics over long horizons. A very recent line of work (e.g., IoT-LLM (An et al., 2025)) highlights that LLMs can be augmented with IoT signals to reason about physical-world tasks. However, benchmarks used in these studies are short-term and simplified, which risks overestimating model capability. In contrast, RIoT, with its multi-site coverage, semantic tagging, and real-world variability, offers an opportunity to design more challenging and realistic benchmarks. We see this as an exciting direction for future work.

## 6 CONCLUSION

In this study, we introduce RIoT, a large-scale dataset that enables access to real-world, multi-modal, and spatially grounded IoT data. As such, it addresses the current scarcity of comprehensive real-world IoT data in research. We hope that it will serve as a foundation for future developemnts of systems that move beyond predefined commands toward more natural, context-aware interactions, pushing the boundaries of what are currently considered truly intelligent IoT.

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

## A    REPRODUCIBILITY STATEMENT

All data and code, preprocessing scripts, visualization scripts, and experiments are provided in an anonymous repository, which is mentioned in the *Introduction* section of the document. For convenience, we also append it here: https://anonymous.4open.science/r/RIoT-9798.

## B    LLM USAGE

LLMs were used solely as assistive tools for grammar and spelling checks during the preparation of this manuscript. The authors take full responsibility for the content of this work.

