# OpenReview forum: "RIoT: A Large-Scale Real-World IoT Dataset"
_ICLR.cc/2026/Conference — Submitted to ICLR 2026_

### Official Review · Reviewer_oz5e · 2025-10-25

**Soundness:** 3
**Presentation:** 2
**Contribution:** 3
**Rating:** 4
**Confidence:** 3

**Summary:**

The manuscript presents RIoT, a real-world Internet of Things dataset collected from six building deployments over a cumulative six-year period. The dataset includes 228 sensors, more than 10 million measurements, and multiple modalities (temperature, humidity, CO₂, VOCs, energy, occupancy, etc.), semantically tagged using the Project Haystack standard. The authors demonstrate two use cases: anomaly detection and HVAC optimisation, and release the dataset and code publicly.

**Strengths:**

1.	RIOT collects and proposes a larger-scale and more diverse IOT dataset, which is multi-site, multi-modal, and long-term. It is far larger and richer than most existing IoT datasets.
2.	Use of Haystack makes the dataset spatial and context-aware, improving usability for building analytics.
3.	This is a timely contribution that addresses a known bottleneck in IoT and smart-building research: the lack of realistic, long-duration datasets.

**Weaknesses:**

- Although RIoT is an impressive data collection effort, the evaluation is minimal. The experiments are mostly illustrative, as they do not establish RIoT as a benchmark dataset with standardised tasks or the most recent baselines. For example, it demonstrates use cases on anomaly detection and simple threshold-based occupancy modelling, without establishing standardised benchmarks (e.g., forecasting, imputation, causal inference).

- All data comes from building environments, but the broader IoT domains include other domains as well such as industrial, urban mobility, or wearable systems.

- The collected data would be more reliable if there were quantitative analyses, such as coverage density across sites.

**Questions:**

No further questions

**Details Of Ethics Concerns:**

The privacy of data in the manuscript is protected and mentioned in Section 2.

---

### Official Review · Reviewer_QE4y · 2025-10-26

**Soundness:** 3
**Presentation:** 3
**Contribution:** 3
**Rating:** 6
**Confidence:** 5

**Summary:**

The paper introduces RIoT, a multi-site, multi-modal real-world IoT dataset intended to move research beyond synthetic or lab-scale settings. RIoT aggregates over 10 M measurements from 228 sensors deployed across six buildings, with site/floor/room spatial tags (Project Haystack) and an on-change and 15-min polling acquisition policy via LoRaWAN gateways. The sites run for 8–18 months each (82 site-months total ≈ 6.8 “cumulative” years). The paper also provides two illustrative use cases: (1) anomaly detection (firmware outliers; rate-of-change anomalies) and (2) HVAC optimization via CO₂/TVOC-based occupancy proxies, including a back-of-the-envelope energy-savings estimate. Dataset/code are linked in an anonymous repository.

**Strengths:**

1. Originality: First publicly available, spatially annotated, on-change IoT dataset at multi-building scale.
2. Quality: Well-documented collection process; clear sensor metadata; anonymization carefully handled.
3. Clarity: Figures show accumulation curves, coverage, and measurement distributions effectively.
4. Significance: Opens the door to robust, reproducible IoT research beyond synthetic or small-lab data.

**Weaknesses:**

1. Limited anomaly-detection baselines: The paper evaluates anomaly detection using only three lightweight baselines—Z-score, SH-ESD, and a simple LSTM (AUROC 0.83–0.91 for rate-of-change anomalies). These are appropriate for demonstrating dataset utility but do not represent state-of-the-art multivariate or deep temporal models (e.g., OmniAnomaly, TranAD, MSCRED). Including such baselines, or at least discussing their relevance, would strengthen the empirical section and showcase RIoT’s potential for complex temporal reasoning.
2. Minor text inconsistencies: The paper includes small typographical and phrasing issues — e.g., “higer” (L118), “Firmare” (L328), and a potential ambiguity between “six-year period” in the abstract and “82 site-months (≈ 6.8 years cumulative)” in Table 2. Clarifying and correcting these would polish the presentation.
3. Release details: The dataset’s license, data-use terms, and standardized splits (e.g., cross-site or temporal) are unspecified and should be clarified for reproducibility.
4. Ethics transparency: Although anonymization (≤20 km spatial perturbation) is described, including a short governance statement about site consent and responsible use would further reassure downstream users.
5. Limited scope of demonstrated use cases: The demonstrated use cases (anomaly detection and HVAC optimization) are relatively simple given the dataset’s scale and diversity. Incorporating more impactful or complex applications, e.g., multi-sensor fusion, cross-building generalization, or predictive control would better demonstrate the richness and broader research potential of RIoT.

**Questions:**

1. What license will govern the dataset’s final release (e.g., CC-BY-4.0)?
2. Will official train/validation/test splits (e.g., cross-site or temporal) be provided for benchmarking?
3.  How can users link anonymized sites to weather data while preserving privacy?
4.  Are there any labeled anomalies or quality flags beyond firmware outliers?
5.  Any plan to expand anomaly benchmarks to multivariate/time-series SOTA methods?

---

### Official Review · Reviewer_ik7n · 2025-11-01

**Soundness:** 3
**Presentation:** 4
**Contribution:** 2
**Rating:** 2
**Confidence:** 5

**Summary:**

The authors present RIoT, a dataset of IoT sensor measurements collected from six building deployments over 18 months, comprising over 10 million measurements from 228 sensors across multiple modalities (HVAC, energy, air quality). The dataset includes semantic and spatial metadata using the Haystack tagging scheme. Two use-cases are demonstrated: (1) firmware-induced anomaly detection, and (2) HVAC optimization through occupancy modeling using CO$_2$ and TVOC as proxies.

**Strengths:**

1. Real-world data at massive scale. This dataset represents a genuine data collection effort spanning six building deployments with diverse sensor types.
2. The use of Haystack tagging to provide site-floor-room hierarchy and semantic sensor types (tempSensor, CO2Sensor, etc.) is more comprehensive than many existing IoT datasets that lack contextual metadata.
3. The dataset includes diverse modalities (temperature, humidity, CO2, TVOC, energy consumption, air quality) from the same deployments, enabling potential cross-modal analysis.
4. The dataset includes real-world imperfections (sensor downtime, firmware bugs, missing values) that are important for developing robust systems, rather than presenting only clean, curated data.
5. The on-change policy (reporting when values change by >1 unit) combined with 15-minute periodic polling provides finer temporal granularity than many datasets with fixed hourly sampling.

**Weaknesses:**

1. The paper provides raw data but defines no specific representation-learning problems, evaluation metrics, or community benchmarks.
2. 10 million measurements do not enable qualitatively new research directions compared to existing smaller datasets (BDGP2, CU-BEMS). Scale increases coverage but not representational complexity.
3. Only 490 firmware anomalies are explicitly labeled (0.027% of CO2 data). Occupancy is a heuristic proxy, not ground truth.  To truly enable novelty understanding richer semantic labels are necssary.  For instance, this dataset offers no intervention logs, control states, maintenance records, or human behavior annotations. Provides “where” and “what” but missing “why” and “who.”
4. Causal inference is practically impossible. Claims about “smart buildings” and “closing the loop” require causal reasoning, but the dataset provides only observational data with massive unmeasured confounding (e.g., environmental factors, human behavior, control algorithms, building state). No identification strategy proposed.
5. Trivial use-cases. Detecting CO2 = 65,535 ppm is threshold checking with perfect separability, not an interesting ML problem. Rate-of-change anomaly detection is standard time-series work. Neither demonstrates novel capabilities enabled by this dataset.
6. Large gap between claims (“intelligent systems,” “robust deployable solutions,” “closing the sense-actuate loop”) and actual demonstrations (basic anomaly detection on observational data). Steps 7-8 in Figure 5 remain missing despite being highlighted.
7. No demonstration of what research questions require this dataset’s characteristics versus existing resources. Missing engagement with literature on causal inference, graph neural networks for sensor networks, or building energy modeling with uncertainty quantification.
8. Occupancy proxy (CO2/TVOC thresholds) never validated against ground truth. No analysis of how 20km geolocation perturbation affects utility of external weather data. Baseline anomaly detection results (Table 5) lack context about base rates and cost asymmetries.

**Questions:**

1. What specific representation-learning problems does this dataset enable that existing datasets (BDGP2,
CU-BEMS, smaller IoT datasets with similar spatial structure) do not?
2. Without intervention logs or control state annotations, how do you propose researchers use this dataset
for closing the sense-actuate loop as prominently claimed?
3. The HVAC optimization analysis makes strong assumptions about energy consumption and comfort
that are never validated. Can you provide actual experimental evidence that your proposed policy
works?
4. Given that occupancy labels are heuristic proxies, have you validated them against ground truth (e.g.,
camera counts, calendar data, PIR readings where available)?
5. What is the identification strategy for causal inference in this observational data with massive unmea-
sured confounding?

**Details Of Ethics Concerns:**

The dataset appears to have been anonymized, with geographic obfuscation and removal of direct identifiers. However, given the presence of fine-grained temporal data and spatial structure, residual risks of behavioral or location-based re-identification possibly remain. The authors might want to clarify whether a GDPR-compliant impact assessment or equivalent ethical review was conducted and specify the data-sharing terms to ensure responsible use.

---

### Official Review · Reviewer_N3nT · 2025-11-08

**Soundness:** 2
**Presentation:** 2
**Contribution:** 2
**Rating:** 2
**Confidence:** 4

**Summary:**

This paper introduces RIoT, a large-scale, real-world Internet of Things (IoT) dataset collected from six building deployments over a cumulative period of approximately six years. The dataset includes over 10 million sensor measurements from more than 200 devices, spanning multiple modalities such as HVAC activity, air quality, temperature, and energy consumption. Each sensor is spatially tagged using the Haystack semantic standard, enabling detailed contextual modeling of physical environments. The paper presents two demonstration use cases: (1) anomaly detection in firmware readings and rate-of-change patterns across indoor/outdoor contexts, and (2) HVAC optimization through occupancy modeling based on CO₂ and TVOC levels. The work positions RIoT as a foundational benchmark for realistic, multi-modal IoT research, addressing the chronic scarcity of publicly available, long-term, real-world datasets in the field.

**Strengths:**

- RIoT includes six diverse sites, 228 sensors, and over 10 million measurements—orders of magnitude larger than typical IoT benchmarks.
- Adoption of the Haystack tagging standard provides rare, structured context for sensor interrelationships.
- Use cases on anomaly detection and HVAC optimization demonstrate real-world applicability and motivate downstream research.

**Weaknesses:**

- Demonstration experiments are basic; deeper machine learning benchmarks or multi-task evaluations (forecasting, graph modeling, etc.) are missing.
- Focused only on building IoT; lacks diversity across other IoT domains (e.g., automotive, industrial, wearable).
- The paper does not establish standardized splits, baselines, or evaluation metrics for future comparative work.
- While cumulative duration is impressive, per-site recording periods (mostly ~18 months) could be more explicitly defined for reproducibility.
- Some sections (e.g., related work, motivation) are overly descriptive and could benefit from clearer emphasis on novel aspects.

**Questions:**

- Could the authors provide detailed benchmark splits and baseline models to facilitate standardized comparisons across methods (e.g., for forecasting, anomaly detection, or control)?
- How consistent are sensor calibrations across sites and vendors—are cross-site comparisons statistically valid?
- Could the authors discuss plans for extending RIoT to other IoT domains (e.g., transportation, wearables) to generalize beyond building automation?
- How were privacy constraints balanced against the need for spatial fidelity after the 20 km perturbation of geolocation data?
- Are there any tools or APIs planned for easier dataset access and exploration?

**Details Of Ethics Concerns:**

No ethics concerns.

---

### Meta-Review · Area_Chair_5vep · 2026-01-07

**Summary:**

reviewers gave scores of 2,2,4,6, with main weaknesses being:

The paper provides raw data but defines no specific representation-learning problems, evaluation metrics, or community benchmarks. The demonstration experiments are basic; deeper machine learning benchmarks or multi-task evaluations (forecasting, graph modeling, etc.) are missing.

10 million measurements do not enable qualitatively new research directions compared to existing smaller datasets (BDGP2, CU-BEMS). Scale increases coverage but not representational complexity, and lacks diversity across other IoT domains (e.g., automotive, industrial, wearable).

Other issues regarding lack of complete data labeling, lack of defined tasks such as causal inference, use cases being rather simple and not requiring deep machine learning or feature representation.

**Reviewer Concerns:**

The authors did not submit a rebuttal so these concerns were not addressed

**Reviewer Scores:**

The authors did not submit a rebuttal so these concerns were not addressed

---

### Decision · Program_Chairs · 2026-01-26

Reject